# Effects of Short-Term Gluten-Free Diet on Cardiovascular Biomarkers and Quality of Life in Healthy Individuals: A Prospective Interventional Study

**DOI:** 10.3390/nu16142265

**Published:** 2024-07-13

**Authors:** Simon Lange, Simeon Tsohataridis, Niklas Boland, Lisa Ngo, Omar Hahad, Thomas Münzel, Philipp Wild, Andreas Daiber, Detlef Schuppan, Philipp Lurz, Karin Keppeler, Sebastian Steven

**Affiliations:** 1Department of Cardiology, University Medical Center, Johannes Gutenberg University, 55131 Mainz, Germanysimeon.tsohataridis@unimedizin-mainz.de (S.T.); omar.hahad@unimedizin-mainz.de (O.H.); tmuenzel@uni-mainz.de (T.M.); daiber@uni-mainz.de (A.D.); philipp.lurz@unimedizin-mainz.de (P.L.);; 2German Center for Cardiovascular Research (DZHK), Partner Site Rhine-Main, 55131 Mainz, Germany; 3Department of Preventive Cardiology and Medical Prevention, University Medical Center, Johannes Gutenberg University, 55131 Mainz, Germany; philipp.wild@uni-mainz.de; 4Institute of Translational Immunology, University Medical Center, Johannes Gutenberg University, 55131 Mainz, Germany; detlef.schuppan@unimedizin-mainz.de; 5Department of Cardiology, Goethe University Frankfurt, 60596 Main, Germany

**Keywords:** gluten, endothelial function, vascular inflammation, quality of life, flow-mediated dilation, plasma proteomics

## Abstract

Introduction: The exposome concept includes nutrition as it significantly influences human health, impacting the onset and progression of diseases. Gluten-containing wheat products are an essential source of energy for the world’s population. However, a rising number of non-celiac healthy individuals tend to reduce or completely avoid gluten-containing cereals for health reasons. Aim and Methods: This prospective interventional human study aimed to investigate whether short-term gluten avoidance improves cardiovascular endpoints and quality of life (QoL) in healthy volunteers. A cohort of 27 participants followed a strict gluten-free diet (GFD) for four weeks. Endothelial function measured by flow-mediated vasodilation (FMD), blood testing, plasma proteomics (Olink^®^) and QoL as measured by the World Health Organisation Quality-of-Life (WHOQOL) survey were investigated. Results: GFD resulted in decreased leucocyte count and C-reactive protein levels along with a trend of reduced inflammation biomarkers determined by plasma proteomics. A positive trend indicated improvement in FMD, whereas other cardiovascular endpoints remained unchanged. In addition, no improvement in QoL was observed. Conclusion: In healthy individuals, a short-term GFD demonstrated anti-inflammatory effects but did not result in overall cardiovascular improvement or enhanced quality of life.

## 1. Introduction

Within the exposome concept, the significance of daily nutrition emerges as a pivotal element contributing to both sustained well-being [1] and the progression and management of a range of diseases. In addition to exposomic risk factors such as mental stress, noise, or exposure to harmful chemicals, nutrition has gained interest within the last years [2,3]. Especially in western societies, gluten-containing grains such as wheat, barley, and rye are a major source for dietary carbohydrates and protein [4,5]. Besides well-known sensitivities and allergies, gluten-containing cereal products are also involved in diseases such as celiac disease (CeD). The latter is an autoimmune enteropathy triggered by dietary gluten in genetically predisposed individuals. Patients suffering from CeD develop a broad range of symptoms including abdominal pain, loss of weight, diarrhoea, and malabsorption [6,7]. It has also been shown that untreated CeD patients suffer from endothelial dysfunction, an early risk marker for the development of an atherosclerotic cardiovascular disease such as ischaemic heart disease [8,9]. To date, the only known therapy for CeD is the life-long avoidance of any gluten-containing foods and beverages. Besides gluten, which makes up 75–85% of storage protein in wheat [10], there are further compounds found in these grains, which are currently studied for their (auto-)immune modulations in non-celiac murine and human models. Recent studies investigated the immune modulatory effects of alpha-amylase trypsin inhibitors (ATIs), which are present in all gluten-containing grains. ATIs are a group of 19 peptides that are part of the plants’ herbivore resistance mechanisms where their contents were increased during the last decades due to crop breeding effects. Since ATIs are resistant to high temperatures and salt concentrations, even refined products made from ATI-containing grains such as bread, pizza, or pasta contain active forms of these peptides in a high concentration [11,12]. Via the pro-inflammatory Toll-like receptor 4 (TLR4) pathway, exposure to active ATIs can lead to intestinal inflammation and deterioration of pre-existing inflammatory diseases such as non-alcoholic fatty liver disease and colitis [13,14,15]. Another group of compounds found in gluten-containing cereals, fermentable oligo-, di-, monosaccharides, and polyols (FODMAPs) is known for their modulatory effects on the composition and diversity of the gut’s microbiome [16,17,18]. ATIs, FODMAPs, and gluten are also being investigated as potential triggers for symptoms associated with non-celiac wheat sensitivity and irritable bowel disease [19,20,21].

These findings, along with efforts to enhance health and well-being, contribute to an increasing number of healthy individuals without diagnosed celiac disease avoiding gluten in their daily nutrition [22,23]. Thus, the present study aims to investigate whether a short-term avoidance of gluten can benefit cardiovascular markers, including inflammation and vascular endothelial function, and quality of life in healthy subjects.

## 2. Materials and Methods

### 2.1. Study Procedures

We conducted a prospective interventional study in which the influence of a gluten-free diet on various health outcomes was investigated (Figure 1). Recruitment took place in the area of Mainz via postings in public places and social media from July 2022 until January 2023. Potential participants between 18 and 60 years old introduced themselves to the corresponding physician at the University Medical Center of the Johannes Gutenberg University Mainz, who decided if participants were suitable for study participation. Exclusion criteria were pregnancy, smoking, cardiovascular, and autoimmune diseases including celiac disease or any drug treatment within the last two months (oral contraceptives excluded). Regular consumption of gluten-containing products was also defined as inclusion criteria. The subjects underwent an initial examination (Visit I), including the assessment of vascular function, blood collection, and evaluation of quality of life through a questionnaire. Subsequently, each subject received comprehensive instruction on adhering to a gluten-free diet. Participants were thoroughly informed about gluten-containing beverages and foods as well as contamination with gluten when eating out. In addition, hidden sources of gluten (fast food, beverages, processed foods) and gluten-free alternatives were discussed. After the 30 min briefing, the subjects were also provided with an information sheet. Coming from a conventional, gluten-containing diet, all participants followed a strict gluten-free diet for four weeks during this study. Subjects documented all meals and beverages consumed in a daily paper-pencil diary that was reviewed for adherence to gluten abstinence. For precise documentation, participants were instructed to fill in the diary right after consuming a meal. Beside food and beverages well-being, performance capability and quality of sleep were ranked with points spanning from 1 (very bad) to 5 (very good). During the four-week GFD period, the study team contacted the subjects once a week to ensure compliance with the gluten-free diet and evaluate their well-being. During Visit I and II (after four weeks of gluten-free diet) vascular function, blood collection, and completion of a quality-of-life questionnaire (WHOQOL-BREF), were performed.

### 2.2. Ethical Approval and Subjects

All human data were collected in accordance with the declaration of Helsinki, and ethical approval was granted by the Landesärztekammer Rheinland-Pfalz [Mainz, Germany; permit number: 2022-16369_1]. Written consent was received from all included individuals. Twenty-seven healthy participants (twenty-two females, five males; mean age: 26.9 ± 4.9 years; BMI 22.6 ± 2.6 kg/m^2^) were enrolled in this study. Before obtaining consent, each volunteer underwent a thorough informational session conducted by the responsible physicians, providing comprehensive details about gluten-containing foods and available alternatives.

### 2.3. Functional, Biochemical, and Clinical Chemistry Parameters

At Visit I and Visit II, participants were scheduled for appointments in the morning between 8 and 10 a.m. and were required to be fasting since the evening before. In case of non-fasting, the appointment was rescheduled for the next day at the same time. Initially, the flow-mediated vasodilation (FMD) was determined as follows. The non-invasive measurement assessing the FMD of the brachial artery uses high-resolution ultrasound and is performed by trained technicians [24]. The brachial artery’s diameter is determined both before and after an increase in shear stress induced by reactive hyperaemia. The sphygmomanometer cuff is placed proximal to the brachial artery and inflated to 200 mmHg for 5 min. Upon cuff release, the reactive, flow-dependent dilation of the brachial artery is documented. The extent of dilation predominantly reflects endothelial function and consequently, the bioavailability of vascular nitric oxide (NO). Ultrasonic-assisted FMD measurements were performed on the brachial artery of the right arm as shown before [25,26]. After measurement of FMD, blood pressure and heart rate were determined followed by blood sampling. The analysis of the blood samples has been performed at the Institute of Clinical Chemistry and Laboratory Medicine of the University Medical Center Mainz, Germany, following standard protocols. Interleukin 6 (IL6) levels in serum were analysed by electrochemiluminescence immunoassay, C-reactive protein concentrations in plasma were tested by immunoturbidimetry and leucocyte number was determined by automated cell count of EDTA blood samples. Plasma levels of adrenalin, noradrenalin, and dopamine were tested as doublets using TECAN^®^ TriCat ELISA assay (IBL International Corp., Toronto, ON, Canada) following the manufacturer’s protocol. Additionally, all participants filled out the 26 questions of the WHOQOL-BREF survey at both visits. Results were analysed for physical/psychological health, social relationships, and environment as intended by the WHO [27], resulting in a score for every domain between 0 (very bad) and 100 (very good).

### 2.4. Plasma Proteomics

Thawed EDTA plasma samples of both visits from ten randomly selected participants (gender-matched: five male, five female) were analysed for their plasma proteome using the Olink^®^ Target 96 Inflammation (95302) and Olink^®^ Target 96 Cardiovascular II (95500) panels. Experimental implementation was performed by the Institute of Preventive Cardiology and Medical Prevention at the University Medical Center Mainz. Results are presented as normalised protein expression (NPX) on a log2 scale. Analysis and visualisation were accomplished by Olink^®^ Insights Stat Analysis application (Olink^®^, Uppsala, Sweden) and GraphPad Prism v10.1.0 (Boston, MA, USA).

### 2.5. Statistical Analysis

Normal distribution and significance tests were conducted using GraphPad Prism v10.1.0 (Boston, MA, USA). Gaussian normal distribution was tested by Anderson–Darling test, Shapiro–Wilk test, and Kolmogorov–Smirnov test. Avoiding Type I errors by performing a Bonferroni correction, *p* < 0.0166 was set as significance level here. To assess the impact of the gluten-free diet on the investigated parameters, a paired Student’s *t*-test with a 95% confidence interval was employed. All tests were two-sided and statistical significance was considered achieved when *p* < 0.05. For this level of significance, a statistical power of 80% was considered.

## 3. Results

Twenty-seven enrolled subjects (twenty-two female, five male; mean age 26.9 ± 4.9 years; body mass index (BMI) 22.6 ± 2.6 kg/m^2^) finished the study after four weeks of GFD. Four initial participants dropped out before or during GFD due to personal reasons. Dropping out was caused by time conflicts or discomfort during the first FMD measurement. Only data from the 27 subjects absolving both appointments successfully were included. Recruitment of male subjects turned out very difficult. Even a prolonged recruitment phase just for male subjects led not to a gender-matched cohort. For all completed subjects, compliance was ensured by weekly calls and the analysis of the nutrition diary. Apart from gluten avoidance, participants were advised to continue all behaviours as before the GFD (e.g., physical activity, sleep rhythm, alcohol consumption). All data regarding blood pressure measurements, blood analysis, hormone plasma levels, and FMD measurements are listed in Table 1.

### 3.1. Functional Parameters

Following the GFD, no significant changes regarding body mass index, systolic, and diastolic blood pressure were observed. Heart rate was increased without any significance after GFD (*p* = 0.1046). Endothelial function measured by flow-mediated dilation was 9.45% before and 11.61% (both normalised for baseline diameter) after GFD (Table 1). This trend to an improved vascular function was not significant (*p* = 0.0897). FMD at baseline or after recovery was not different between both groups.

### 3.2. Biochemical and Clinical Chemistry Parameters

The complete blood count demonstrated a significant decline in leucocytes (Table 1) after gluten avoidance (*p* = 0.0135), while levels of erythrocytes (*p* = 0.7182), haemoglobin (*p* = 0.9195), and thrombocytes (*p* = 0.0505) differed without significance. There was no change regarding the investigated levels of vitamin B12, folic acid, total cholesterol, LDL-, and HDL-cholesterol in accordance with triglycerides observed following the GFD. The inflammatory marker C-reactive protein (CRP) was decreased by trend after GFD (*p* = 0.0689), whereas IL6 showed no changes (*p* = 0.7976) (Table 1). There was also no evidence for changes of the hepatic markers glutamic oxaloacetic transaminase (GOT) (*p* = 0.5871), glutamic pyruvic transaminase (GPT) (*p* = 0.4663), glucose, HBA1c, or homocysteine after four weeks of gluten-free diet. ELISA analysis of plasmatic hormone levels showed no alteration of adrenalin (*p* = 0.4556), noradrenalin (*p* = 0.6385), and dopamine (*p* = 0.9920).

### 3.3. Plasma Proteomics

Plasma proteins has been analysed via Olink^®^ proximity extension assay technique. A panel regarding inflammatory markers (Olink^®^ Target 96 Inflammation) and a second panel focussing on cardiovascular health (Olink^®^ Target 96 Cardiovascular II) have been used. Plasma samples drawn at both visits were analysed on the same plate. Results of plasma proteomics are visualised in Figure 2 and Appendix A.

No significant changes towards a reduced inflammatory phenotype after four weeks of GFD were observed. The proto-oncogene tyrosine-protein kinase SRC was significantly reduced. However, no other protein target was significantly regulated after four weeks of gluten-free diet. A mild trend of reduced pro-inflammatory plasma proteins (e.g., TNF, CCL3, CD40) and chemokines (e.g., IL6, IL8, IL18) was observed post-GFD.

### 3.4. Quality of Life Parameters

Subjective well-being pre- and post-GFD was investigated by the WHOQOL-BREF survey. Evaluation of the survey revealed no significant variations in terms of physical (*p* = 0.9529) and psychological health (*p* = 0.6671), social relationships (*p* = 0.4430), and environment (*p* = 0.1955). Furthermore, there has been no changes in sleep quality, performance capability, or well-being, which has been evaluated every week. Results of this investigation are displayed as weekly means in Figure 3.

## 4. Discussion

With the present study, we aimed to investigate effects of a four-week GFD on functional cardiovascular, biochemical/clinical chemistry parameters, and QoL in healthy subjects. The hypothesis that a short-term GFD might lead to significant changes in parameters such as vascular function, blood pressure, inflammation, or quality of life was not confirmed. The study demonstrates limited short-term impact on endothelial function and no impact on QoL parameters. The observed reduction in leukocyte count (5.24 ± 1.18 vs. 5.00 ± 1.25, *p* = 0.0135) within whole blood and reduced levels of SRC from plasma proteomics have been the sole statistically significant changes noted after participants followed a GFD for four weeks. Along with that, a trend towards reduced CRP in line with results of performed plasma proteomics and by trend improved flow-mediated dilation as a read-out for endothelial function was demonstrated.

Active CeD is associated with an increased risk of cardiovascular disease, which was suggested by several clinical-epidemiological studies [19,28,29]. Gluten-related disorders like CeD have a low prevalence, affecting around 1% of western populations. Despite this rarity, there has been a substantial surge in adopting gluten-free lifestyles and consuming gluten-free products in the past 30 years, with over $15.5 billion spent on retail sales of gluten-free foods in 2016 [30]. The gluten-free trend is driven by factors such as media coverage, aggressive marketing, and reports stating health benefits. People may adopt gluten-free diets for various reasons, including the perceived improvement of symptoms and the belief that it promotes a healthy lifestyle. A study within the NutriNet-Santé cohort aimed to characterize sociodemographic, behavioural, and dietary profiles of individuals avoiding gluten. Out of 20,456 participants, 10.31% reported avoiding gluten, particularly women, older individuals, non-smokers, and those with lower educational levels. These participants exhibited a dietary pattern associated with healthier food choices, including higher fruit and vegetable consumption and lower intake of dairy, salty/sweet/fatty foods, and alcohol [22]. These individuals tend to have an overall healthier diet. However, a correlation between a GFD and improved cardiovascular health is not substantiated. While some evidence supports gluten avoidance for certain gastrointestinal symptoms, the overall high-quality evidence for benefits beyond immune-mediated responses to gluten is limited [18,30]. The prevailing negative media portrayal of wheat and gluten has influenced public perception, leading some individuals to consider diets containing gluten as potentially harmful to health. Contrary to this belief, scientific research suggests otherwise, highlighting wheat’s role as a valuable source of whole grains. This dietary component is known for its health benefits, suggesting that a wheat-containing diet may support health rather than pose a risk. This context calls for a more differentiated understanding of the nutritional impact of wheat on the human diet.

In a study involving over 100,000 participants without celiac disease, Lebwohl et al. discovered no link between prolonged dietary gluten consumption and the risk of coronary heart disease [31]. Intriguingly, the findings suggested that non-celiac individuals who adopt a gluten-free diet may inadvertently elevate their risk of heart disease by potentially reducing their intake of whole grains. Numerous studies have established a positive correlation between the consumption of whole grains, including wheat, and improved health outcomes. For instance, groups with the highest whole grain intake (2–3 servings daily) exhibited significantly lower rates of heart disease, stroke, the development of type 2 diabetes, and overall mortality compared to those consuming the lowest amounts (less than 2 servings daily) [32,33,34]. Moreover, gluten may serve as a prebiotic, nourishing the beneficial bacteria within our bodies. Arabinoxylan oligosaccharide, a prebiotic carbohydrate derived from wheat bran, has demonstrated the ability to stimulate the activity of bifidobacteria in the colon. These bacteria play a crucial role in maintaining a healthy human gut, and disruptions in their quantity or activity have been associated with gastrointestinal disorders, including inflammatory bowel disease, colorectal cancer, and irritable bowel syndrome [35,36]. On the other hand, beneficial gluten avoidance was observed in non-celiac disease like dermatitis herpetiformis [37], rheumatoid arthritis [38], and diabetes mellitus type I disease onset [39].

In our investigated healthy cohort, we did not observe any modulation of vitamin B12 or folic acid levels. Due to decreasing microvilli blunting, these micronutrients are often positively influenced when diet changes towards GFD are performed in CeD patients [6]. In concordance with previous long-term gluten-avoidance studies [31], no improvements on parameters impacting cardiovascular outcome such as vascular function, lipid profile, blood glucose, or blood pressure were observed following short-term gluten avoidance in this study. This might be the case due to dietary changes towards higher contents of salt, saturated fatty acids, and sugar. Clinical trials investigating GFD composition found a significant increase in sugar, fat, and salt along with imbalanced uptake of micronutrients when GFD is applied in CeD patients as well as in healthy subjects [40,41,42,43]. Effects of these nutritional deterioration might interfere with beneficial effects of GFD regarding vascular health resulting in non-significant alterations of FMD and further cardiovascular marker. To decipher this interplay of beneficial and adverse effects found in the NutriNet-Santé cohort and the study presented here, a more detailed investigation of changes in diet and lifestyle by GFD is required.

We did not identify any negative implications of the gluten-free diet as it was found in long-term studies due to lower intake of whole grain products, which are known for their protective effects on the cardiovascular system via high levels of fibre [31,44]. Furthermore, the occurring effect on observed leukocyte levels cannot be accounted to the gluten abstinence only. Strict gluten-free diet usually goes along with a strong decrease in consumption of ATIs and FODMAPs. Regarding FODMAPs, a variety of studies already demonstrated the conducive effect of a diet low on FODMAPs in patients suffering from irritable bowel disease [16,17]. The avoidance or reduction of FODMAPs resulted in attenuation of symptoms, healthier microbiome constitution, and a subjective quality of life [45,46,47]. Dietary ATIs are known for their pro-inflammatory properties via the TLR4 pathway in the gut, triggering the endotoxin-linked immune response [48]. Exposure to ATIs was investigated by studies in humanised mouse models and human-derived cell culture. They revealed increased secretion of plasmatic inflammation marker (Il6, Il8, TNFa), as well as activation of macrophages and leucocytes infiltrating distant tissues via CD4^+^ T cells as well as fuelling pre-existing auto-immune disease [13,14,15,49]. Analysis of plasma proteomics revealed a trend to reduced levels of chemokines and inflammatory biomarker downstream of the mentioned TLR4 pathway following four weeks of GFD. Hence, the observed alteration was not significant. These findings can give a hint for further studies to investigate the inflammatory properties of dietary ATIs in humans. Therefore, further studies regarding this group of pro-inflammatory compounds are required to decode which cereal constituent is the main trigger for inflammation and which coherent groups of patients can profit from dietary changes.

In summary, the present data demonstrate that short-term GFD does not have an effect on functional cardiovascular, biochemical/clinical chemistry parameters, or quality of life. On the other hand, it can also be stated that a short-term gluten-free diet is not harmful in healthy subject in a four-week period. While the study provides valuable insights into the potential effects of a four-week GFD, several limitations must be considered when interpreting the results. The study involved a relatively small sample size of 27 participants, with only 5 male subjects, which is likely underpowered. This limited representation may affect the generalizability of the findings to a broader population. A larger and more balanced sample would enhance the external validity of the study. The four-week duration of the gluten-free diet may be insufficient to capture long-term changes in the measured parameters, such as systemic inflammation. Longer intervention periods could reveal more sustained effects on endothelial function, hormonal levels, and other health outcomes. Furthermore, the inclusion of a completely untreated control group would be sufficient to exclude potential side effects caused by the intervention itself or time effects during the observed period. Although the WHOQOL-BREF survey is a widely used tool, relying solely on subjective self-assessment for evaluating the participants’ quality of life may introduce bias. Combining subjective measures with objective indicators could offer a more comprehensive assessment. The study did not thoroughly investigate participants’ dietary habits beyond gluten avoidance. Therefore, more detailed documentation of consumed meals (amount in gram, volume of beverages, and manufacturer of processed foods) during GFD is needed as well as the same documentation in the weeks before and after GFD. This would allow conclusions on behavioural changes due to a performed short-term GFD. Furthermore, providing a sufficient amount of gluten-free substitute products (e.g., pasta, bread, and flour) could contribute to a higher degree of standardisation in such a trial. A completely predetermined diet with known macro- and micronutrients as well as defined mealtimes could provide more representative suitability in such a study design avoiding inaccuracies regarding the frequency and amount of food consumed. Understanding potential variations in nutrient intake or compensatory dietary changes could contribute to a more comprehensive interpretation of the observed outcomes.

## 5. Conclusions

In conclusion, despite the valuable findings presented in this study, addressing these limitations in future research will contribute to a more robust understanding of the effects of gluten-free diets on health parameters. Additional studies with larger and more diverse samples, longer intervention periods, and comprehensive analytical approaches are warranted to validate and extend the current observations.

## Figures and Tables

**Figure 1 nutrients-16-02265-f001:**
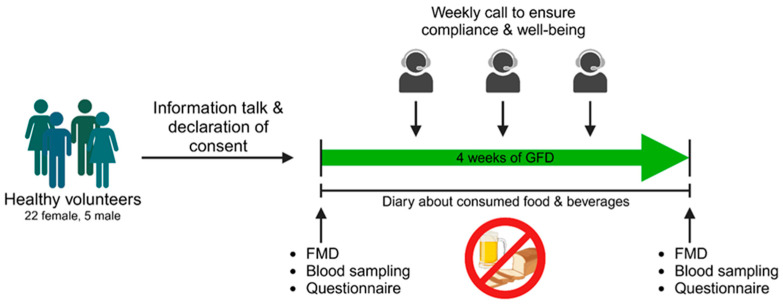
Study procedures. After an informational session and the signing of consent forms, each volunteer underwent a standardized procedure, commencing with flow-mediated vasodilation (FMD) measurements, blood sampling, and the completion of the WHOQOL-BREF questionnaire. This process was conducted before initiating a four-week gluten-free diet (GFD). Adherence to the diet was monitored through weekly calls and the maintenance of a nutrition diary. Following the gluten-free diet, the same parameters were examined as in the pre-diet phase. Designed with BioRender.

**Figure 2 nutrients-16-02265-f002:**
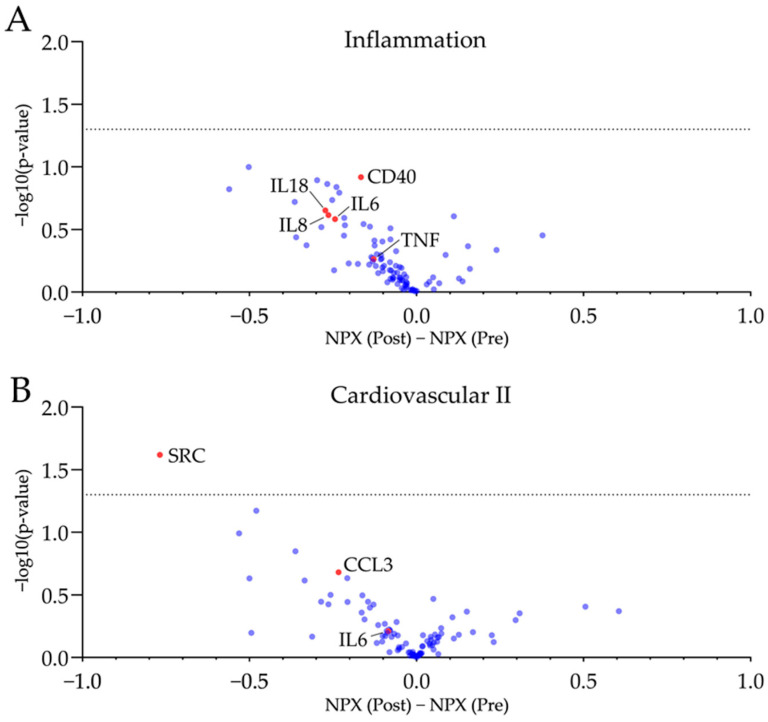
Plasma proteomics revealed no significant changes in plasma proteins associated with inflammation and cardiovascular health after four weeks of GFD. Volcano plots of the used inflammatory (**A**) and cardiovascular panels (**B**) display a mild trend towards reduced pro-inflammatory markers with just SRC being reduced significantly (−log10(*p*-value) > 1.3). Further, inflammation--associated biomarker and chemokines are reduced by trend following four weeks of GFD. NPX, normalized protein expression. n = 10, unpaired *t*-test, dotted line is −log10(*p*-value) = 1.3, equals *p*-value = 0.05.

**Figure 3 nutrients-16-02265-f003:**
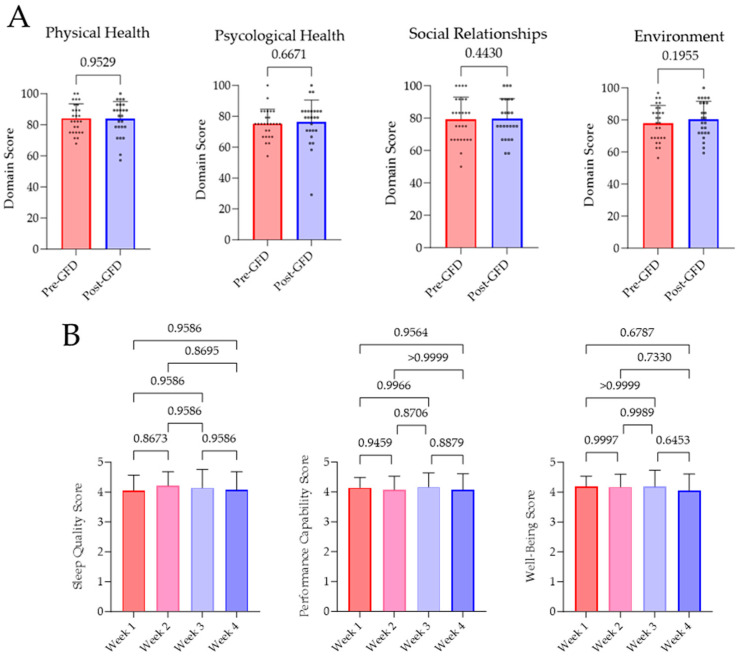
Results of the WHOQOL-BREF survey and the daily perception of sleep quality, performance capability, and well-being. (**A**) Questionnaires were answered by all subjects before and after four weeks of GFD. Analysis and evaluation of the results were performed as intended by the WHO [27]. (**B**) Participants rated their subjective perception on sleep quality, performance capability, and well-being from 1 (very bad) to 5 (very good) on every day of GFD. In a week-wise comparison, no significant changes were determined.

**Table 1 nutrients-16-02265-t001:** Physiological parameters from blood pressure measurements, blood analysis, ELISA, and FMD measurements prior and after four weeks of GFD. n = 27.

	Pre GFD (t0)	Post GFD (t1)	Significance Level
	Mean ± SD	Mean ± SD	n.s. *p* > 0.05 * *p* < 0.05
Age (years)	26.97 ± 4.99		
Gender	22 female. 5 male		
Height (cm)	170.74 ± 7.40		
BMI (kg/m^2^)	22.60 ± 2.60	22.20 ± 2.33	n.s., *p* = 0.3019
Heart Rate (bpm)	64.07 ± 7.24	68.44 ± 9.72	n.s., *p* = 0.1046
**Blood pressure**			
Systolic BP (mmHg)	121.52 ± 9.60	124.00 ± 10.30	n.s., *p* = 0.3118
Diastolic BP (mmHg)	77.55 ± 6.95	78.00 ± 7.63	n.s., *p* = 0.7966
**Complete Blood Count**			
Erythrocytes (/pl)	4.45 ± 0.38	4.45 ± 0.35	n.s., *p* = 0.7182
Haematocrit (%)	39.08 ± 2.81	39.25 ± 2.37	n.s., *p* = 0.8470
Haemoglobin (g/L)	13.13 ± 1.01	13.21 ± 0.85	n.s., *p* = 0.9195
MCV (fl)	87.95 ± 3.27	88.52 ± 3.12	n.s., *p* = 0.6196
MCH (pg)	29.55 ± 1.33	29.78 ± 1.20	n.s., *p* = 0.4450
MCHC (g/dL)	33.58 ± 0.84	33.66 ± 0.83	n.s., *p* = 0.7030
Thrombocytes (/nL)	224.89 ± 50.33	210.65 ± 41.22	n.s., *p* = 0.0505
Leucocytes (/nL)	5.24 ± 1.18	5.00 ± 1.25	*, *p* = 0.0135
**Serum vitamins**			
Folic acid (ng/mL)	6.40 ± 2.36	7.06 ± 2.07	n.s., *p* = 0.2449
Vitamin B12 (ng/L)	361.28 ± 103.93	382.80 ± 99.04	n.s., *p* = 0.6470
**Lipid profile**			
Triglycerides (mg/dL)	70.03 ± 22.18	62.42 ± 15.81	n.s., *p* = 0.7973
Cholesterol (total) (mg/dL)	172.62 ± 29.39	168.12 ± 28.75	n.s., *p* = 0.9871
HDL (mg/dL)	62.52 ± 8.88	62.62 ± 9.83	n.s., *p* = 0.4032
LDL (mg/dL)	96.00 ± 21.59	92.96 ± 21.35	n.s., *p* = 0.7073
LDL/HDL-ratio	1.56 ± 0.33	1.52 ± 0.36	n.s., *p* = 0.3512
**Hepatic profile**			
Albumin (g/L)	43.45 ± 2.43	42.88 ± 2.59	n.s., *p* = 0.2947
GOT (U/L)	23.83 ± 5.38	23.31 ± 4.22	n.s., *p* = 0.5871
GPT (U/L)	19.31 ± 5.10	22.12 ± 7.91	n.s., *p* = 0.4663
**Inflammatory profile**			
CRP (mg/L)	0.89 ± 0.66	0.62 ± 0.31	n.s., *p* = 0.0689
Interleukin 6 (ng/L)	2.56 ± 0.61	2.54 ± 0.84	n.s., *p* = 0.7976
**Hormone profile**			
Adrenalin (pg/mL)	66.23 ± 35.65	64.55 ± 34.72	n.s., *p* = 0.4556
Noradrenalin (pg/mL)	473.13 ± 189.77	517.48 ± 278.03	n.s., *p* = 0.6385
Dopamine (pg/mL)	62.50 ± 28.75	63.26 ± 28.71	n.s., *p* = 0.9220
**Vasodilation profile**			
FMD (% of baseline diameter)	9.45 ± 5.39	11.61 ± 3.41	n.s., *p* = 0.0897
FMD baseline	3.51 ± 0.51	3.49 ± 0.45	n.s., *p* = 0.6241
FMD recovery	3.74 ± 0.47	3.86 ± 0.54	n.s., *p* = 0.2509
**Others**			
Glucose (mg/dL)	83.62 ± 5.61	85.35 ± 5.42	n.s., *p* = 0.4890
HBA1C (%)	5.14 ± 0.17	5.08 ± 0.19	n.s., *p* = 0.2970
Homocysteine (µmol/L)	8.72 ± 2.17	8.99 ± 2.12	n.s., *p* = 0.9368
TSH (mU/L)	1.46 ± 0.55	1.42 ± 0.65	n.s., *p* = 0.9358

GFD, gluten-free diet; BMI, body mass index; BP, blood pressure; MCV, mean corpuscular volume; MCH, mean corpuscular hemoglobin; MCHC, mean corpuscular hemoglobin concentration; HDL, high density lipoprotein; LDL, low density lipoprotein; GOT, glutamic oxaloacetic transaminase; GPT, glutamic pyruvic transaminase; CRP, C-reactive protein; FMD, flow-mediated vasodilation; HBA1C, glycated haemoglobin; TSH, thyroid stimulating hormone.

## Data Availability

The data presented in this study are available on request from the corresponding author due to legal regulations concerning the subjects’ privacy.

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
