# Peer review of "Effects of Short-Term Gluten-Free Diet on Cardiovascular Biomarkers and Quality of Life in Healthy Individuals: A Prospective Interventional Study"

_nutrients, 2024, doi:10.3390/nu16142265_

Round 1
Reviewer 1 Report
Comments and Suggestions for Authors
In this work, the authors have investigated the effect of 4 weeks gluten free diet on several parameters correlated to the endothelial function, cardiovascular and inflammatory biomarkers, and on quality of life in healthy subjects. The manuscript is clear, the results well presented and discussed. However, the work presents many limitations, as correctly stated by authors in the discussion, such as low number of subjects, mainly women, and a too short intervention period. This is probably the reason for the null effects observed on most of the parameters investigated. A statistically significant effect was observed only for the leucocytes count and the proto-oncogene tyrosine-protein kinase SRC. Indeed, the C-reactive protein levels were reduced but the result was not statistically significant. This makes the article of little scientific soundness and little interest to readers.
I have just two suggestion for the manuscript: 1) since the intervention strategy was extremely simple, I think that Figure 1 (even if it is a nice picture) is not necessary for the understanding; 2) Figure 2 (which is an identical repetition of the results reported in Table 1) is also not necessary.
Reviewer 2 Report
Comments and Suggestions for Authors
This paper provides valuable insights into the effects of a short-term gluten-free diet on cardiovascular biomarkers and quality of life in healthy individuals. The study is well-conceived, although several limitations should be addressed in future research. The discussion presents a balanced view of the potential benefits and drawbacks of gluten-free diets. Overall, this manuscript holds the potential for publication in the journal; minor revisions are suggested for further consideration.
1. Comment
In lines 154-155, the authors state, “Thawed EDTA plasma samples of both visits from ten randomly selected participants were analysed for their plasma proteome …” For clarity and to strengthen the study’s methodology, could the authors elaborate on the rationale behind selecting 10 participants for plasma proteomics analysis instead of including all 27 participants? Additionally, how many males were there among the 10 participants? Understanding the reasoning behind this choice will help the audience better assess the representativeness and reliability of the proteomics data.
2. Comment
In the “Statistical Analysis” section, could the authors provide more detail on the statistical method(s) used to correct for Type I errors when conducting multiple statistical tests?
3. Comment
In lines 171-173, the authors state, “Twenty-seven enrolled subjects finished the study after four weeks of GFD. Four initial participants dropped out before or during GFD due to personal reasons.” Does this mean that a total of 31 participants were initially enrolled in the study, with 27 completing it and having their data reported in this manuscript? If so, could the authors add one or two sentences to clarify the participant flow for the audience?
4. Comment
In line 171, the authors state that there were 27 enrolled participants, comprising 22 females and 5 males. Could the authors provide the reasons behind the significant sex imbalance in this study? Did the authors consider halting enrollment of females and continuing enrollment of males to achieve a balanced sex representation in the study?
5. Comment
In Table 1, could the authors include the sample size (n) in the row titles “Pre GFD (t0)” and “Post GFD (t1)”?
6. Comment
Could the authors provide sample size (n) for each plot? Additionally, it would be helpful if the authors specify the interpretation of the bar plots and error bars. For instance, do the bars represent mean values with error bars indicating standard deviations (mean ± SD)?
7. Comment
In lines 190-191, the authors state, “Endothelial function measured by flow-mediated dilation (FMD) was 9.31% before and 11.42% (both normalised for baseline diameter) after GFD (Figure 2).” However, according to Table 1, the FMD values at Pre GFD (t0) and Post GFD (t1) were 9.45% and 11.61%, respectively. Could this be a typographical error?
8. Comment
It appears that the parameters and values depicted in Figure 2 are already detailed in Table 1. Could the authors clarify how Figure 2 enhances the presentation of data in this manuscript? Was the inclusion of Figure 2 primarily intended for data visualization purposes? Please note, my question aims to understand the rationale behind Figure 2's inclusion and does not imply a suggestion for its removal.
9. Comment
In lines 307-308, the authors state, “The observed reduction in leukocyte count (5.24 ± 1.18 vs. 307 5.00 ± 1.25, p=0.00135) within whole blood…”, however, in Table 1, the p-value is 0.0135. Please correct the typographical error.
10. Comment
In this study, participants were permitted to decide their own gluten-free diet (GFD) rather than following a standardized diet provided by the investigators, potentially leading to variations in meal composition (e.g., frequency, amount, food content) among participants. This variability could potentially compromise the study’s rigor. It would be beneficial for the authors to discuss this limitation in the manuscript.
